# Weathering and Material Characterization of ZTO/Ag/ZTO Coatings on Polyethylene Terephthalate Substrates for the Application of Flexible Transparent Conductors

Yu-Han Kao [1], Hung-Shuo Chang [1], Chih-Chieh Wang [2] and Chiao-Chi Lin [1,*]

1   Department of Materials Science and Engineering, Feng Chia University, Taichung 40724, Taiwan
2   Department of Materials and Optoelectronic Science, National Sun Yat-Sen University, Kaohsiung 80424, Taiwan
*   Correspondence: chiaochi@fcu.edu.tw

**Abstract:** Flexible transparent conducting coatings have been adopted in many novel optoelectrical devices for energy-related applications. Laminated coatings composed of oxide/metal/oxide (abbreviated as OMO) layers are promising alternative materials to indium tin oxide (ITO). However, the durability and weatherability of free-standing OMO samples—including laminated OMO coatings and polymeric substrates—affects the performance of the related optoelectrical components and devices. It is necessary to study the degradation mechanisms in terms of optoelectrical and mechanical properties through the weathering tests. In this study, we performed indoor accelerated and outdoor weathering tests on commercial OMO samples composed of ZTO/Ag/ZTO coatings on polyethylene terephthalate (PET) substrates. The effects of environmental stressors such as ultraviolet (UV) radiation, elevated temperature, and mechanical bending on the degradation behaviors of OMO samples were investigated. Comprehensive material characterizations including UV–Vis spectroscopy, four-point probing, tensile tests, and Raman spectroscopy were carried out. The OMO coating was robust against the weathering tests, while the PET substrates underwent embrittlement upon long-term weathering. The embrittled PET substrates consequently impaired the mechanical flexibility and bendability of the OMO coatings. The results of this study provide an in-depth understanding of the durability and weatherability of silver-based OMO flexible transparent conductive materials.

**Keywords:** flexible electronics; transparent conductors; silver; laminated coatings; multiple layers; weathering; durability; polymeric substrate; bending deflection

## 1. Introduction

Silver-based flexible transparent conductive materials have great potential to replace indium tin oxide (ITO) in flexible and printable optoelectrical components and devices. Research and development of key materials promotes the realization of novel flexible optoelectrical components and devices for energy applications such as organic light-emitting diodes, organic photovoltaics, antennas for energy harvesting, and self-powered wearable electronics [1–3]. Recently, the technology of laminated coatings composed of oxide/metal/oxide (abbreviated as OMO) layers has reached commercialization level for mass production and commercial use for the requirements of flexible transparent conductive films. However, the durability and weatherability of the materials—including laminated nanofilms with a transparent conductive metal oxide/silver mesh/metal oxide structure and polymeric substrates—affect their performance [4–9]. The reliability of the fabricated optoelectrical components and devices is accordingly affected; therefore, the viability of widespread commercialization for flexible and printable optoelectronics is impeded. In order to tackle potential shortcomings in device design and fabrication, it is worthwhile to study the degradation mechanisms of OMO flexible transparent conductors in terms of optoelectrical and mechanical properties through weathering tests.

The conduction mechanism of the laminated OMO structure mainly takes place through the conduction of the silver mesh layer [1]. To achieve a 3D growth for silver film deposition that follows the Volmer–Weber growth mode, a minimum thickness of 5–10 nm has to be attained. When the thickness of the silver mesh layer is thicker than the minimum thickness, the sheet resistance value is lower, but so is the optical transmittance in the visible light range. However, in the thickness region of a few tens to a few hundred nanometers, the situations are reversed—i.e., when the silver mesh layer is thinner, the sheet resistance value is significantly higher, but the optical transmittance in the visible light range reaches a plateau due to the photon resonance confinement effect generated by the nanostructure of the silver mesh and the metal oxide layers. Moreover, the compressive residual stress increases due to the confinement effect of cohesion force which, in turn, affects the lattice constant and valence band energy level [4]. Therefore, the optical transmittance in the visible light region exhibits a behavior of decreasing if the film thickness of silver is thinner than a minimum value about 10 nm or thicker than a critical value of about 100 nm. The conductive properties of the OMO structures are relatively stable, while the optical transmittance properties strongly depend on the microstructure and morphology of the OMO materials.

The influence of microstructure and microscopic residual stress on the optoelectrical properties of OMO structures is of great importance. Sarma et al. analyzed the microscopic grain stress via the X-ray diffraction (XRD) $\sin^2 \psi$ method. It is known that the residual stress has a direct impact on the optoelectrical properties of the OMO structure [4]. Winkler et al. proved the phenomenon of surface diffusion and migration of nano-atomic agglomerations on the surface of ZTO (zinc tin oxide, i.e., Sn-doped ZnO)/Ag/ZTO through atomic force microscopy (AFM) observations immediately after the coatings had been deposited [5]. Therefore, the influence of the residual stress on the laminated OMO structure can be significant, and the importance of pre-stabilizing before use for the flexible transparent conductors of OMO materials has been emphasized.

The results of the material stability testing of OMO samples at 85 °C/85% high temperature and high humidity (i.e., damp heat (DH) conditions) in the literature show that SnO has better structural stability than other conductive metal oxides. As for the metal oxide materials, Behrendt et al. experimentally confirmed that doping with Sn is beneficial for improving the stability of the metal oxide layer [6]. On the other hand, as for the metal layer, the research results of Jeong et al. show that doping with oxygen atoms in the silver interstitial sites can reduce the free energy of the interface for the silver mesh layer [7], greatly improving the ability of dewetting and material stability, and inhibiting thermal degradation. In addition, because ultraviolet (UV) light can easily decompose and recombine oxygen molecules in the environment to form stable oxygen atoms and ozone [8], UV light is inevitably an environmental factor that impairs the flexible transparent conductors of the laminated OMO structures and polymeric substrates [9]. However, the weatherability and stability problems caused by the effects of UV exposure at elevated temperatures have not been reported in any literature so far.

Polyethylene terephthalate (PET) is a common substrate material for flexible transparent conductive films due to its mature industrial manufacture, high visible light transmittance, and low cost. Research on PET's weatherability has been developed for decades. In principle, the degradation mechanism caused by UV irradiation is classified into Norrish type I and Norrish type II. The formation of free radicals has an impact on molecular weight, crystallinity, mechanical properties, and optoelectrical properties [10]. When coupled with high temperature, humidity, chemical pollutants, and the application of stressors such as electrical stress, the outdoor degradation behavior of PET is variable and complex. Moreover, the substrate surface of commercial PET products used in outdoor applications is usually coated with a transparent inorganic layer (e.g., SiOx or $Al_2O_3$, usually several nanometers in thickness) that hardens and blocks moisture. However, due to different formulations from various PET suppliers, in recent years there have still been a number of updated research results in this field [11,12]. For the accelerated weathering testing of PET,

the UVA band of UV radiation (wavelength around 340 nm) is a suitable simulator if the artificial irradiance intensity level is moderate.

The relationship between the bending mechanical behavior and electrical conductivity of metal oxide conductive film/PET substrate systems has been well studied both theoretically and experimentally [13–17]. In recent years, due to the rise of emerging technologies such as large-area sputtering of metal thin films and silver nanowires (AgNWs), the research attention has gradually migrated to focus on novel nanomaterials such as OMO and oxide/AgNWs/oxide structures on PET substrates. Research fields such as optoelectrical properties, coating material design, and interface engineering have been integrated with the investigation of basic bending mechanical properties. However, the literature on materials' reliability- and weatherability-related studies connecting to the above research fields and end applications is still lacking.

The mechanism of the decrease in the conductivity of flexible transparent conductive films caused by cyclic bending is mainly attributed to the generation and accumulation of bending stress and strain-induced defects. The shear lag occurs at the material interfaces of the conductive film, the OMO coating, and the substrate when bent. The normal stress at the interface of the film/substrate also presents discontinuity. In addition, the conductive layers of ceramic–metal coatings and the PET polymeric substrate present a significant mismatch in Young's moduli, so that bending can generate a huge number of defects in the OMO coatings and start to propagate and accumulate, resulting in cracks and, hence, affecting the conductivity. Continuously increasing the number of bending cycles promotes the initiation and growth of cracks, and increases the crack density, eventually causing saturated crack density and failure [15–17].

In this study, we performed accelerated and outdoor weathering tests for the investigation of the material degradation behaviors of OMO coatings on PET substrates. Firstly, the materials and microstructures of the OMO samples were analyzed. Then, after the accelerated and the outdoor weathering tests, the material characterization of the free-standing OMO samples was carried out, and the correlation between the indoor and outdoor degradation behaviors was investigated. Finally, combining several environmental factors—including temperature, UVA, and subsequent cyclic bending—the effects of UV light exposure and mechanical bending on the mechano-optoelectrical behaviors of OMO coatings and polymeric substrates were studied. The root causes of material degradation and failure were investigated.

## 2. Experimental Methods

### 2.1. OMO Samples and Weathering Tests

Commercial OMO samples purchased from ConvergEver Inc., Ltd. (Taoyuan City, Taiwan) were studied. The optoelectrical properties of the as-received OMO samples (in a size of 10 cm $\times$ 10 cm) were characterized using the four-point method and a UV–Vis spectrophotometer. The OMO samples were composed of nanolaminated coatings fabricated by room-temperature sputtering technology on PET substrates. The laminated OMO structure was ZTO/Ag/ZTO, providing an original sheet resistance of 4.7 $\pm$ 0.1 ohm/sq (Figure 1a) and corresponding transmittance of 83 $\pm$ 2% at a wavelength of 550 nm in the visible-light region (Figure 2). The PET substrate was 125 μm in thickness, while the thickness of the silver film was about 100 nm. Pre-aging at 150 °C for 30 min was carried out to stabilize the initial geometric dimensions of the OMO samples prior to the weathering tests [5]. It should be noted that the pre-aging process did not alter the electrical properties of the OMO coatings due to the hard coatings on the PET surface [11], while the optical properties slightly changed (Figure 2)—as discussed and explained later in this paper—due to the aging effect of the PET substrate.

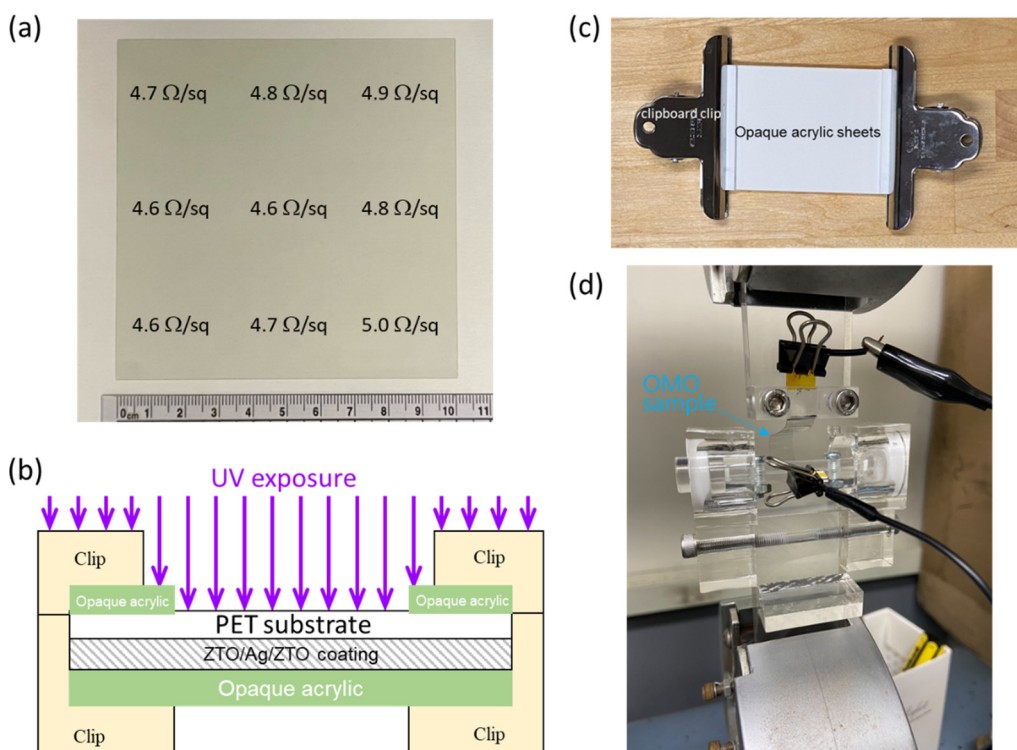

**Figure 1.** (**a**) A photo of the OMO sample with the corresponding measured sheet resistance values. (**b**) The experimental setup of the weathering tests. (**c**) A photo of the sample holder for the weathering tests. (**d**) Real-time resistance measurement under cyclic bending.

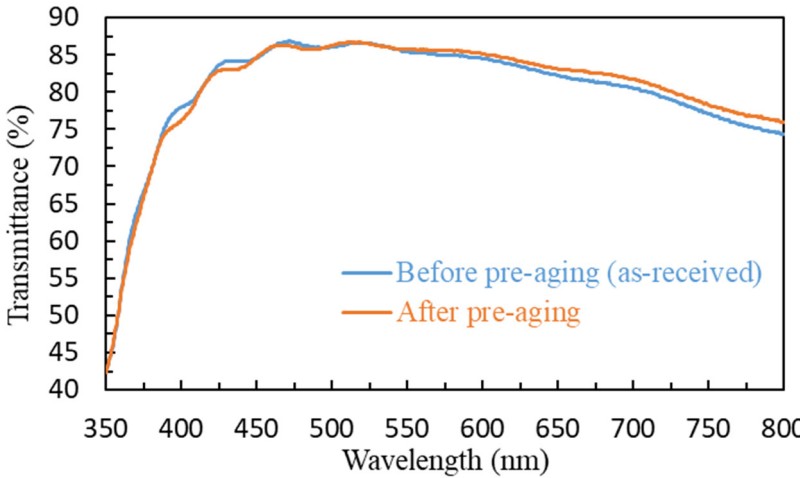

**Figure 2.** UV–Vis spectra of the OMO sample as received and after pre-aging.

The detailed exposure setup for the weathering tests is shown in Figure 1b,c. Square samples with a size of 5 cm × 5 cm and rectangular strips with two sizes of 5 cm × 0.5 cm and 10 cm × 1.0 cm were cut out from the sheets of the OMO samples purchased from the manufacturer. All sample-cutting procedures were conducted using a rotary trimmer (KW-trio 13060, Changhua County, Taiwan) to diminish edge defects and make the samples as nick-free as possible [18]. We paid special attention to keep the long axis of the long strip sample parallel to the transverse direction—rather than the working direction—of the OMO sample's processing direction. The square and rectangular OMO samples were used to measure the optoelectrical properties and the mechanical properties through the tensile/cyclic bending tests, respectively.

For weathering tests, we put the OMO sides of the samples down and clamped them with a clipboard clip, as shown in Figure 1. The indoor accelerated weathering was performed in a UVA exposure tester (UD-403S, JobHo Technology Co. Ltd., Taichung City, Taiwan) with the parameters of UVA irradiance of 0.5 W/m$^2$·nm (at 340 nm) at 75 °C to simulate the sun at noon. The indoor accelerated weathering test, which represents continuous sun, was carried out for 23 days. On the 5th, 10th, 13th, 17th, 20th, and 23rd days of exposure, some samples were taken out, and then their optoelectrical and mechanical properties were characterized. The field test of outdoor weathering was performed using an under-glass exposure box conforming to ASTM G24 and G201 standards. The exposure site of the field tests was located on the campus of the Feng Chia University in Taiwan [9,19]. The outdoor weathering test was carried out for 92 days starting from 15 July 2021. On the 20th, 50th, and 92nd days of outdoor weathering, some samples were taken back for selective characterization of their materials in terms of their optoelectrical and mechanical properties. Free-standing PET substrates without OMO coatings were also investigated when necessary.

## 2.2. Materials' Characterization

For optoelectrical properties, we used a Hitachi UV–visible (UV–Vis) spectrophotometer (U-5100, Tokyo, Japan) and a resistance measurement multimeter (LRS4-TG1, KeithLink Technology Co., Ltd., New Taipei City, Taiwan) consisting of a four-point probe stage to measure the transmittance spectra and sheet resistance (Rsq) of the 5 cm × 5 cm OMO samples, respectively. The degradation behaviors of the PET substrates were analyzed using a Raman spectrometer (Ocean Insight, QEPRO-532 nm, Orlando, FL, USA) with a laser wavelength of 532 nm. The Raman measurements were carried out with the parameters of power of 200 mW and integration time of 5 s. There were at least two OMO samples characterized for each exposure time, and each OMO sample was characterized at two or more locations. The characterized locations were away from the edges of the 5 cm × 5 cm OMO samples. Tensile tests were carried out using a universal testing machine (Chun Yen Testing Machines Co., Ltd., CY-6102, Taichung City, Taiwan) with a gauge length of 3 cm for the 5 cm × 0.5 cm rectangular OMO samples. The crosshead speed was 5 mm/min during the tensile testing.

We used a scanning electron microscope (SEM, S-4800, Hitachi, Tokyo, Japan) to observe the surface morphology of the OMO coatings. A nanolayer of platinum was sputtered before the SEM observation in order to avoid the charge accumulation of high-energy electrons on the substrate. The SEM working voltage was 4~10 kV. When necessary, coating surface observation was also performed using an optical microscope (OM). An X-ray photoelectron spectroscope (XPS, ULVAC-PHI, PHI 5000 VersaProbe, Kanagawa, Japan) was used for qualitative chemical composition analysis. The light source of the XPS was Kα of 1486.6 eV from an aluminum target, and the working voltage was 15 kV.

## 2.3. Real-Time Resistance Measurement under Cyclic Bending

Subsequently, after the weathering tests, cyclic bending tests were performed on the 10 cm × 1.0 cm free-standing OMO samples. Using a combined system of the resistance measurement multimeter and the universal testing machine with a servo control function, real-time resistance measurements were conducted during cyclic bending testing. The testing setup is shown in Figure 1d. We designed custom clamps made of acrylics so that the 10 cm × 1.0 cm samples could be bent when the movement of one clamp was reciprocating back and forth. Meanwhile, electrical connections using alligator clips to both ends of the OMO sample and the digital multimeter facilitated real-time measurement to record the resistance values of the OMO samples under cyclic bending. When the OMO sample was bent, the conductive coating surface was the convex side (i.e., the PET substrate surface was the concave side) at the center of the sample. The minimum radius of curvature of the bent OMO samples was kept at 1.0 cm. The traveling speed of the cyclically reciprocating movement of the acrylic clamp was 300 mm/min, resulting in a bending cycle of about

40 s per cycle. The failure criterion for the OMO samples under cyclic bending was set as a threefold increase in resistance compared to the original resistance value, and the cyclic bending test was terminated when this criterion was met. OMO coatings of failed samples were inspected selectively through SEM for failure analysis.

## 3. Results and Discussion

### 3.1. Unaged OMO Samples

Figure 3 shows the composition of the OMO coatings characterized by XPS. The signals of zinc, tin, oxygen, carbon, and silicon can be clearly seen, while the signal of silver (Figure 3b) is very weak due to the laminated structure of the OMO and the mesh-like format of the silver layer [1]. The metal oxide layer is Sn-doped ZnO (zinc tin oxide (ZTO)), as indicated in Figure 3c,d. The signals of carbon and silicon might come from the protective layer at the surface of the OMO coatings and the hardened layer at the coating–substrate interface.

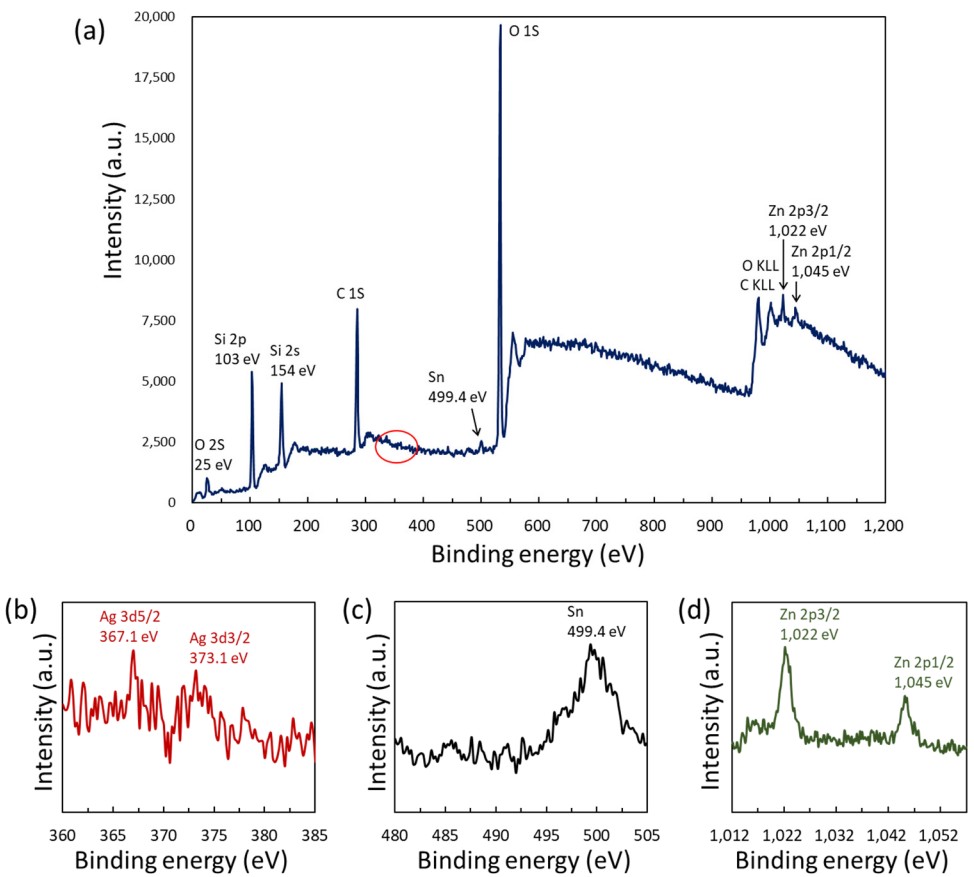

**Figure 3.** XPS spectra of ZTO/Ag/ZTO coatings: (**a**) full spectrum (the red circle indicates the barely seen signal from silver); (**b**) fine-scan spectrum for Ag; (**c**) fine-scan spectrum for Sn; (**d**) fine-scan spectrum for Zn. Peak assignment is shown accordingly.

The Raman spectrum of the flexible transparent conductive film of the laminated OMO coatings on the PET substrate is shown in Figure 4. It is clear to see that the spectrum is a typical PET Raman spectrum. The vibration modes of the PET molecular structure corresponding to each Raman peak position are assigned in Table 1. The main characteristic bands of PET are the stretching of the carbonyl group (C=O) at 1734 cm$^{-1}$ and a set of characteristic bands at 1125 cm$^{-1}$, 1102 cm$^{-1}$, and 1007 cm$^{-1}$. Mixed modes commonly attributed to benzene ring CH include in-plane bending, ethylene glycol C-O stretching, COC and CCO bending, and C-C bond stretching of PET [20,21]. However, there is no

detectable Raman signal for zinc tin oxide due to its weak photonic interaction and the insufficient sampling volume of its nanofilm.

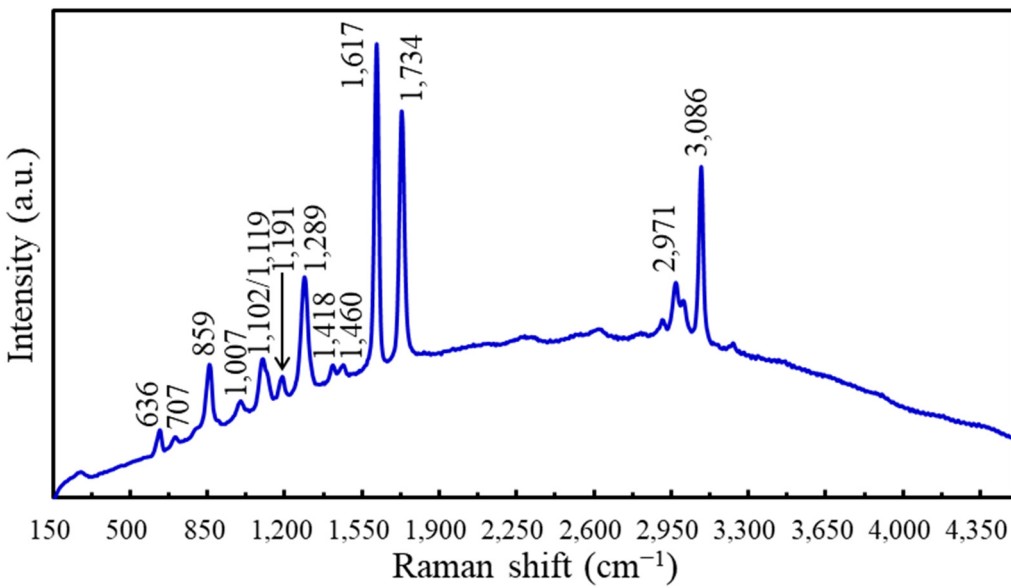

**Figure 4.** Raman spectra of the OMO sample, mainly showing the signal from the PET substrate.

**Table 1.** Band assignment of the Raman spectrum for the PET substrate [20,22,23].

| Raman Shift (cm$^{-1}$) | Assignment |
| --- | --- |
| 636 | Ring C–C–C in-plane bend |
| 707 | Ring C–C–C out-of-plane bend |
| 859 | Ring C–C breathing |
| 1007 | Glycol C–C stretch/O–CH$_2$ stretch/ring torsion |
| 1102 | Ring CH in-plane bend/glycol C–O stretch/COC and CCO bending/C–C stretch |
| 1191 | Ring CH in-plane bend |
| 1289 | Ring-carbonyl stretch/O–C stretch/ring CH in-plane bend |
| 1418 | Ring C–C stretch |
| 1460 | Glycol C–C deformation |
| 1617 | Ring C=C stretch |
| 1734 | C=O stretch |
| 2971 | Amorphous aliphatic CH$_2$ stretching |
| 3086 | Aromatic C–H stretching |

The conductivity measurement indicates that the uniformity of the OMO samples is excellent, and the difference in sheet resistance values between measured locations and between samples is within ±3%. The results of UV–Vis spectroscopy for the OMO samples indicate that the transmittance at a 550 nm wavelength in the visible-light region is 83 ± 2%. According to the above results, it can be concluded that the uniformity and robustness of the laminated OMO coatings are excellent for producing the flexible transparent conductive free-standing film as a commercial product. The substrate is a standard typical PET polymer, which presents excellent initial material properties, including optical transmittance and mechanical flexibility.

### 3.2. Degradation of OMO Coatings and PET Substrates

After accelerated weathering through UVA exposure at 75 °C, the sheet resistance values of the flexible transparent conductive OMO films did not change (as shown in Table 2). Similar results were found from the testing results of outdoor weathering (Table 3). Sheet resistance values increased slightly as the exposure time of the field test extended. The differences in degradation behaviors between indoor and outdoor weathering tests were due to the existence of a thermal cycle. Thermal stress caused by the hot/cold cycle from day to night can gradually impair the adhesion of the laminated OMO structure. Overall, the robust thermal stability of the OMO coating was due to the coexistence of a tin oxide phase in the ZTO layer [6]. Moreover, the coexisting phase of tin oxide densifies the microstructure of the ZTO layer, so that the silver mesh layer can be protected by the ZTO/Ag/ZTO laminated structure against harsh environments, protecting the silver from oxidation and sulfidation.

**Table 2.** Sheet resistance values of OMO samples before and after indoor accelerated weathering tests.

| Indoor Accelerated Weathering | | | | | |
|---|---|---|---|---|---|
| Exposure Time (Day) | 0 | 5 | 10 | 13 | 17 | 23 |
| Rsq ($\Omega$/Square) | $4.7 \pm 0.1$ | $4.8 \pm 0.2$ | $5.3 \pm 0.3$ | $5.0 \pm 0.2$ | $4.9 \pm 0.2$ | $4.9 \pm 0.5$ |

**Table 3.** Sheet resistance values of OMO samples before and after outdoor weathering tests.

| Outdoor Weathering | | | |
|---|---|---|---|
| Exposure Time (Day) | 0 | 20 | 50 | 92 |
| Rsq ($\Omega$/Square) | $4.7 \pm 0.1$ | $6.3 \pm 1.0$ | $7.8 \pm 1.9$ | $9.9 \pm 1.5$ |

The optical properties before and after weathering tests can be seen in Figure 5. There were only slight changes before and after the accelerated and outdoor weathering tests. It can be presumed that the degradation of the PET substrate results in changes to its surface and the interfacial morphologies at the nanometer scale [24]. Therefore, the PET substrate's degradation alters the surface/interface plasmon resonance mode caused by the nanostructure of the silver mesh [25]. Such a PET degradation does not have much influence on the optical transmittance in the visible-light range on a macroscopic scale, as illustrated in Figure 5. On the other hand, the transmittance in the band around a 400 nm wavelength decreases as the exposure time increases, indicating that the yellowing of the PET substrate resulted from both indoor and outdoor weathering tests [22]. Both indoor and outdoor weathering test results showed similar degradation behaviors of OMO samples. We can conclude that the indoor accelerated weathering can effectively simulate the outdoor field tests for the weatherability testing of OMO samples.

To further understand the degradation mechanisms of OMO samples, the analysis results of the Raman spectra of the OMO samples before and after the accelerated aging tests are shown in Figure 6. It can be clearly seen that as the time of the accelerated aging test increases, the fluorescent background signal of the OMO sample increases accordingly. It has been confirmed that UVA exposure at 75 °C causes chain scission of PET polymer molecules, forming free radicals and further generating unsaturated conjugate bonds [22,23]. Therefore, the excitation of Raman spectrometer laser light produces the fluorescence phenomenon from the aged PET substrates. The longer the exposure time of the weathering tests, the more severe the PET substrate's degradation. The fluorescence phenomenon can also be deduced through the evolution of PET yellowing in the near-UV region, as shown in the UV–Vis spectra of Figure 5.

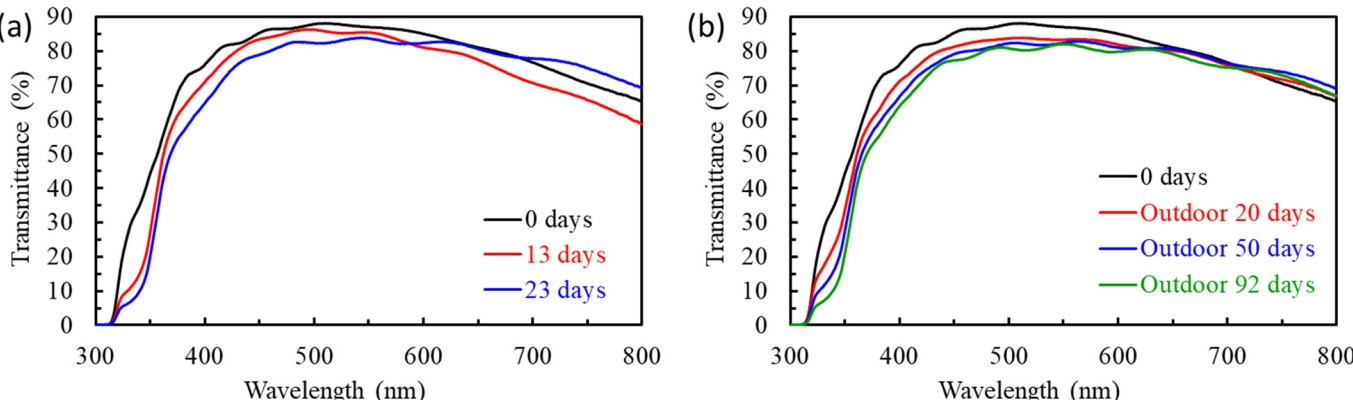

**Figure 5.** UV–Vis spectra showing the optical transmittance of the OMO samples before and after (**a**) indoor accelerated and (**b**) outdoor weathering tests.

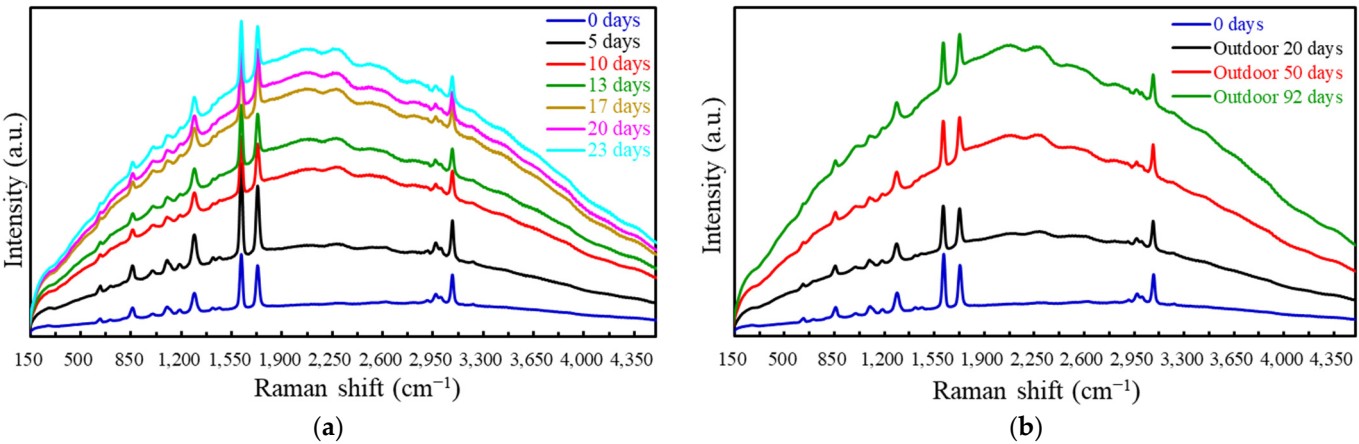

**Figure 6.** Raman spectra of OMO samples before and after the (**a**) indoor accelerated and (**b**) outdoor weathering tests.

Figure 7 shows the mechanical properties of the OMO samples before and after the weathering tests. Figure 7a shows the stress–strain curve of the tensile test results. The mechanical properties of the unaged OMO samples were as follows: Young's modulus of $2.5 \pm 0.2$ GPa, strain at break of $83.1 \pm 3.5\%$, and ultimate tensile strength of $174.7 \pm 5.9$ MPa (Table 4). After 10-day weathering through UVA exposure at 75 °C, the ductility of the OMO samples was significantly reduced, and the mechanical behavior became brittle. The fracture strain was sharply decreased from 83% to 17%, and the fracture stress was also lowered by about half after accelerated weathering for 10 days. When the exposure time of accelerated weathering was more than 13 days, there was no obvious yield point in the stress–strain curve. The outdoor aging resulted in the same behavior of integrity deterioration as the indoor aging.

The mechanical properties of the OMO samples before and after the weathering tests can also be clearly explained through Raman spectroscopy. The 1102 cm$^{-1}$/1119 cm$^{-1}$ peak intensity ratio of the PET Raman spectrum (Figure 4) represents the degree of crosslinking of long-chain molecules form crystalline regions [26,27]; the higher the ratio, the greater the crystallinity of the PET microstructure. When external stress or strain is applied, the crystalline portion provides the capability of long-chain slippage—that is, the highly crystalline PET possesses a higher fracture strain. Figure 7b clearly shows that as the exposure time of the weathering testing increases, the intensity ratio of 1102 cm$^{-1}$/1119 cm$^{-1}$ from the OMO sample's Raman spectrum decreases. After 10 days of accelerated weathering, the peak intensity ratio of 1102 cm$^{-1}$/1119 cm$^{-1}$ reaches a constant value. This result

can be clearly and directly correlated with the changes in mechanical properties over the accelerated weathering time (Table 4).

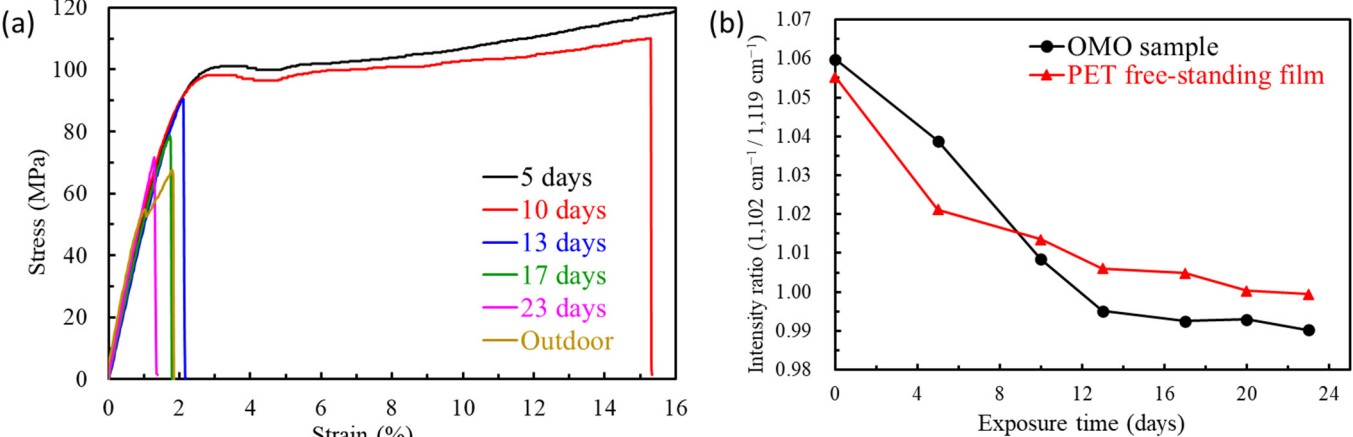

**Figure 7.** (**a**) Stress–strain curves of OMO samples after weathering tests. (**b**) Raman intensity ratio as a function of accelerated weathering time.

**Table 4.** Tensile testing results of OMO samples before and after the weathering tests.

| Exposure Time (Days) | Ultimate Tensile Strength (MPa) | Elongation at Break (%) | Yield Strength (MPa) |
|---|---|---|---|
| 0 | 174.7 ± 5.9 | 83.1 ± 3.5 | 110.2 ± 4.4 |
| 5 | 170.2 ± 10.1 | 66.0 ± 8.5 | 105.9 ± 7.4 |
| 10 | 112.8 ± 6.9 | 17.0 ± 3.1 | 99.0 ± 3.4 |
| 13 | 91.6 ± 6.0 | 2.2 ± 0.1 | N/A |
| 17 | 79.7 ± 7.8 | 1.7 ± 0.1 | N/A |
| 20 | 71.9 ± 7.9 | 1.5 ± 0.1 | N/A |
| 23 | 72.8 ± 6.4 | 1.4 ± 0.2 | N/A |
| Outdoor 50 days | 71.3 ± 9.0 | 2.0 ± 0.2 | N/A |

The Raman spectra of the free-standing PET substrates before and after the accelerated weathering tests were also acquired. The analyzed results shown in Figure 7b illustrate that the change in the 1102 cm$^{-1}$/1119 cm$^{-1}$ intensity ratio as a function of exposure time was the same between the OMO sample and the PET substrate. This illustrates that the loss of mechanical integrity of the OMO samples after weathering is mainly caused by the aging degradation of the PET substrate upon UV exposure [10–12].

Since the mechanism of PET degradation is dominated by UV exposure [10–12], the conversion of the exposure time (in days) of the indoor accelerated weathering to that of the field test provided in Table 5 depicts the correlations between the indoor and outdoor weathering tests in terms of tensile testing results. The UVA exposed the OMO samples to an irradiance of 60 W/m$^2$ for the entire 24 h of 1 d. According to the logged data through the weather station on the exposure site at Feng Chia University, the UV irradiance of natural light is about 40 W/m$^2$. Therefore, 1 d of the UVA testing (UVT) is equivalent to 1.5 d of continuous sun (CS) [28]. To correlate indoor and outdoor aging, an equivalent outdoor condition (EOC) that represents the number of natural days being simulated by the UVT was adopted when there was less than 24 h of sunlight per day in the real world. According to the logged data through the weather station on the exposure site, there is an average of 12 h of sunlight per day in summer, so the EOC conversion factor is 3 (i.e., 1.5 × 24 h/12 h = 3). Therefore, in terms of UV irradiance dosage, 1 d of exposure to the UVT is equivalent to 3 natural days of outdoor exposure to natural light. Hence, the embrittlement behavior of the OMO samples that occurred on the 17th day of indoor

exposure is roughly equivalent to 51 days of outdoor weathering under natural conditions. This is consistent with the data shown in Figure 7 and Table 4.

**Table 5.** Number of days (d) the OMO samples were irradiated with the indoor UVA radiation in the UV testing (UVT), along with the corresponding continuous sun (CS) and equivalent outdoor conditions (EOCs), in days.

| UV Testing (d) | Continuous Sun (d) | Equivalent Outdoor Conditions (d) |
|:---:|:---:|:---:|
| UVT | CS = 1.5 × UVT | EOC = 3 × UVT |
| 5 | 7.5 | 15 |
| 10 | 15 | 30 |
| 13 | 19.5 | 39 |
| 17 | 25.5 | 51 |

*3.3. Bending Mechanical Properties of OMO Samples*

Flexible optoelectrical components and devices are among the crucial technologies for the development of next-generation energy applications and related industry. The electromechanical properties of flexible conductive materials are usually tested and verified by mechanically dynamic cyclic bending and real-time conductivity measurement [14]. As a sequential weathering test procedure in this study, the electromechanical properties of the OMO samples were studied after the accelerated weathering tests under UVA exposure at 75 °C. Additionally, the unaged OMO samples were also tested for real-time electrical and mechanical properties through the bending tests. Figure 8 presents the results of the increase in normalized resistance, $(R - R_0)/R_0$, as a function of cyclic bending cycles for OMO samples with and without accelerated weathering tests. The increase in normalized resistance rose exponentially as a function of cyclic bending cycles for both unaged and aged OMO samples. However, due to embrittlement of the PET substrate after weathering, the aged OMO samples had a higher rate of failure than the unaged ones. The failure rate increased as the exposure time of weathering was extended. The 15-day-aged OMO sample completely lost its ductility, causing and transferring greater stress under the same status of bending deflection, resulting in significant fluctuation of resistance when the OMO sample was under cyclic bending (blue curve in Figure 8).

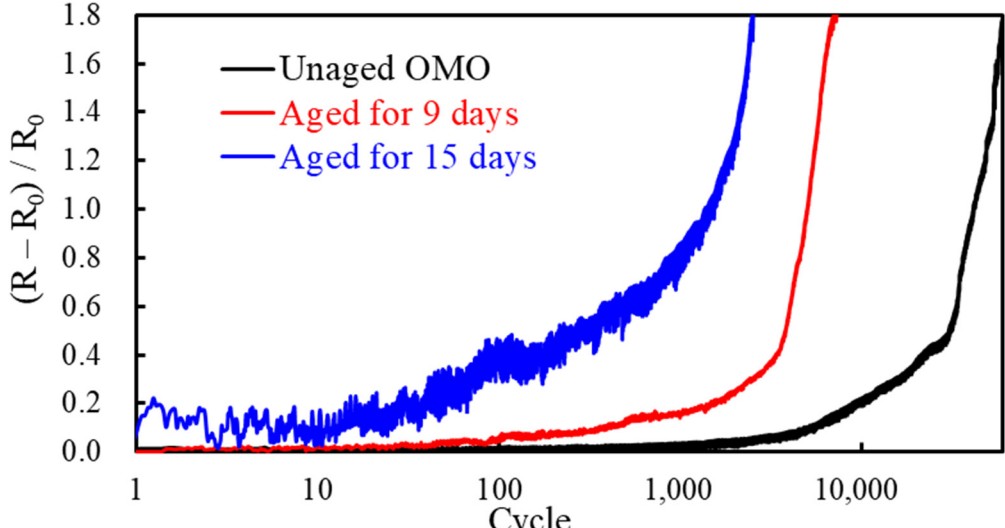

**Figure 8.** Real-time recording results of the increase in normalized resistance as a function of cyclic bending cycles for OMO samples with and without accelerated weathering tests.

The SEM micrographs of the surface morphology of the failed OMO samples after the cyclic bending tests are shown in Figures 9 and 10. The direction of the periodic cracks is always perpendicular to the direction of the applied normal stresses, either in tension or compression. The unaged OMO samples possessed regular, periodic cracks with minor transverse cracks caused by Poisson's effect (Figure 9a). Poisson's effect was significant on the 15-day-aged OMO sample (Figure 9b). The surface cracking behaviors resulting from tensile stress and compressive stress were arguably different. The cracks presenting less buckling caused delamination for the tensile-stress-related cracks (Figure 9b,c), while the compressive-stress-related cracks showed clear buckling, causing delamination at the opening edges of the surface cracks (Figure 10a). The results related to tensile and compressive applied stresses here are consistent with the results in the literature [14,15].

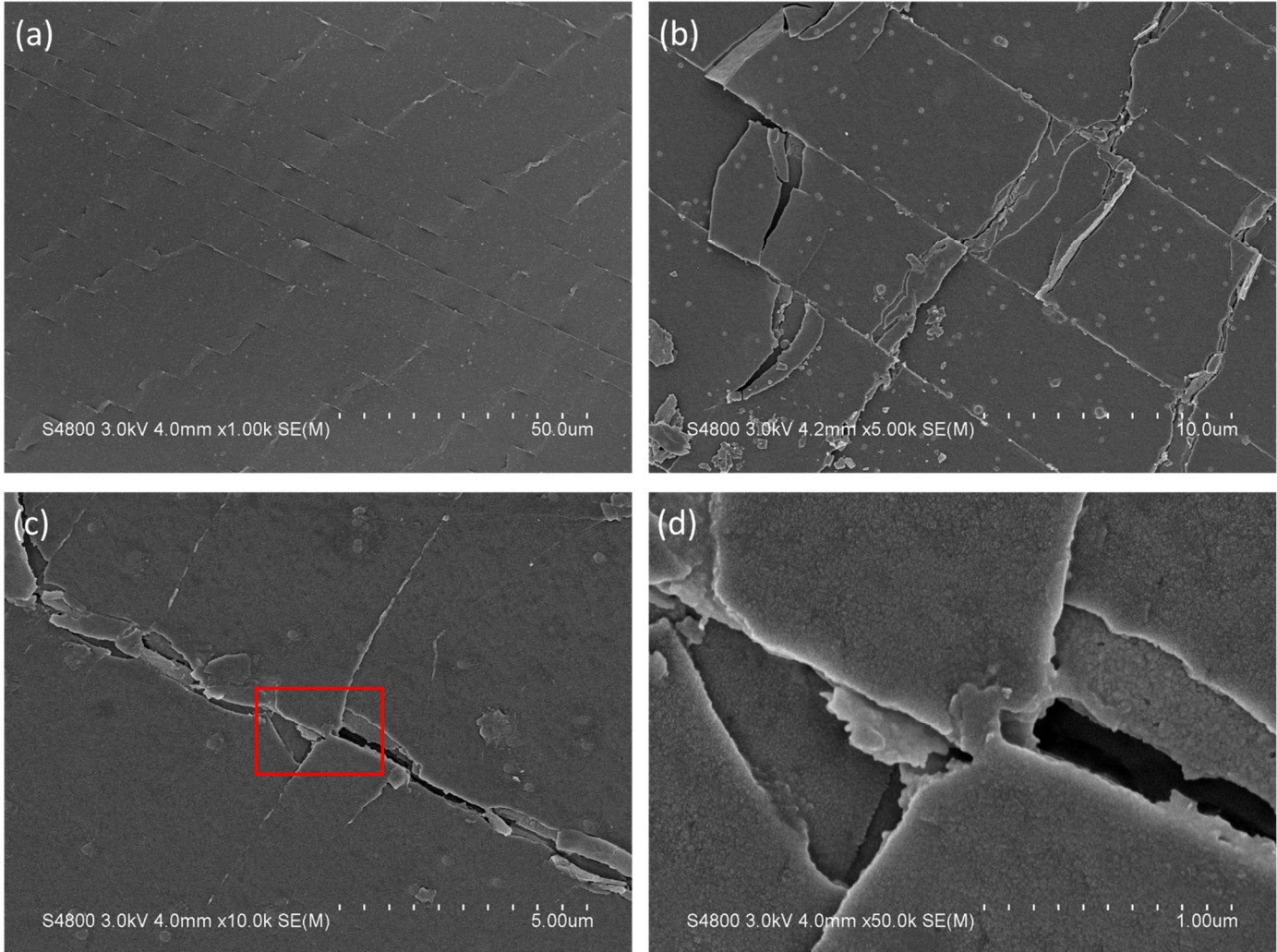

**Figure 9.** SEM images showing the tensile-stress-caused failure of OMO samples after cyclic bending: (**a**) unaged OMO sample; (**b**) low-magnification image of the 15 d aged OMO sample; (**c**) high-magnification image of the 15 d aged OMO sample; (**d**) zoom-in of the red rectangular area in (**c**).

The degraded PET substrate can easily promote delamination due to deteriorated mechanical properties and reduced adhesion of the OMO coatings [29,30]. The results confirm that the dynamic electrical properties under the cyclic bending of the OMO samples are deeply affected by the weathering degradation of the PET substrate. Apart from the surface cracks caused by bending, the microstructure of the OMO coating remained complete, providing good electrical conductivity, albeit along with evenly distributed

surface cracks on the layers (Figures 9d and 10b). Moreover, the more brittle the PET substrate, the higher the crack density that the cyclic bending can produce [30]. Such surface cracking can accelerate the chemical damage to OMO coatings due to the poor protection provided by the oxide layers to the embedded metallic layer, e.g., silver [31]. In addition, in real-world applications, various repetitive mechanical deformations—such as folding, bending, twisting, stretching, sliding, shearing, etc., and their combinations—are very common [32]. Different dynamic deformation parameters and complex alignment between the applied principal stress and sample longitudinal directions result in multiple failure behaviors, with each having characteristic distribution of failure in terms of statistics [15,33]. More in-depth study through statistical analysis of surface cracking behaviors under cyclic bending is in progress in our research group, and will be published in the near future.

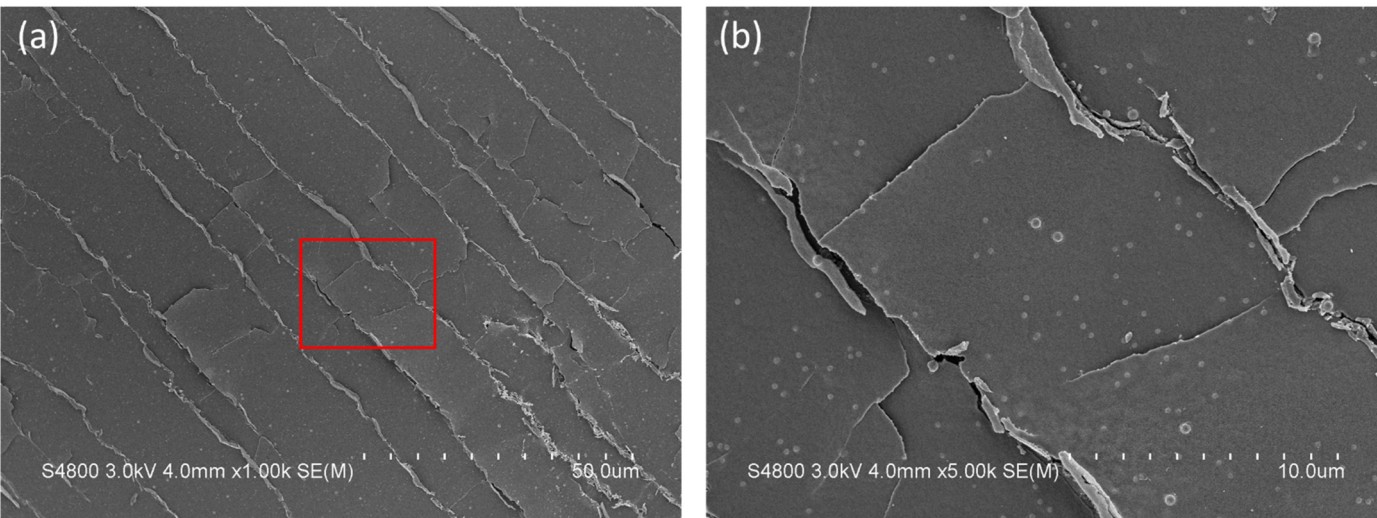

**Figure 10.** SEM images showing the compressive-stress-caused failure of OMO samples after cyclic bending: (**a**) low-magnification image of the 15 d aged OMO sample; (**b**) zoom-in of the red rectangular area in (**a**).

## 4. Conclusions

This study provides in-depth understanding of the weathering degradation and aging behavior of laminated ZTO/Ag/ZTO coatings on polymeric PET substrates as a form of flexible transparent conductive film for energy-related optoelectronics applications. Because of the doping of tin oxide, the microstructure of the ZTO nanolayer is dense, and the stacked ZTO/Ag/ZTO nanostructure can protect the conduction layer of the metallic Ag mesh in harsh environments. However, PET substrates are easily impaired by environmental chemicals (e.g., oxygen, sulfides, and traces of impurities in the process and ambient conditions), moisture, heat, and UVA light. The PET degradation behavior can be accelerated, especially under intensified UV light and elevated temperatures. The aged and degraded PET substrate directly affects the morphology of the interface between it and the conductive OMO coating, but merely affects the plasmonic resonance mode of the nanostructure of the silver mesh. Therefore, the optical transmittance in the visible-light range is not too significantly influenced on a macroscopic scale after long-term weathering. However, yellowing of the PET substrate might cause cosmetic effects on the end products after long-term weathering.

Although several methods have been reported in the literature to improve the stability between the metal oxide layer and the conductive metal layer of OMO coatings, the fundamental problem of PET substrate stability, which is inherently susceptible to weathering, still cannot be completely solved—especially when in operation in a realistic scenario, when the input voltage is applied to the silver nanomesh and its contact pathway. If the aging effect of bending stress and bending strain is added, catastrophic failure will be promoted,

and is more likely to occur as weathering goes on. Therefore, the improvement of the bending integrity and conductivity is essential to improve the lamination design of the OMO coatings, while also choosing suitable substrate materials by taking key performance, weatherability, and expenditure into consideration.

The results of the indoor accelerated weathering tests and the outdoor field tests were consistent, showing the importance of the indoor accelerated weathering tests and the sequential aging tests in verifying the design and manufacture of the materials and structures of the laminated OMO coatings for use as flexible transparent conductive films. The results of this study constitute significant contribution to the improvement of the stability, durability, and weatherability of the laminated OMO coatings against severe outdoor weather for energy applications.

**Author Contributions:** Investigation, formal analysis, Y.-H.K. and H.-S.C.; data curation, visualization, Y.-H.K.; methodology, H.-S.C.; validation, resources, and writing—review and editing, C.-C.W. and C.-C.L.; writing—original draft preparation, Y.-H.K. and C.-C.L.; supervision, C.-C.W. and C.-C.L.; conceptualization, project administration, and funding acquisition, C.-C.L. All authors have read and agreed to the published version of the manuscript.

**Funding:** This research was funded by the Ministry of Science and Technology (MOST), Taiwan, grant number 109-2221-E-035-039 and 111-2221-E-035-051.

**Institutional Review Board Statement:** Not applicable.

**Informed Consent Statement:** Not applicable.

**Data Availability Statement:** All data that support the findings of this study are included within the article.

**Acknowledgments:** The authors thank the Global Research and Industry Alliance (GLORIA) at the Feng Chia University for their help in the maintenance of the outdoor testing site and facilities.

**Conflicts of Interest:** The authors declare no conflict of interest. The funders had no role in the design of the study; in the collection, analyses, or interpretation of data; in the writing of the manuscript, or in the decision to publish the results.

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
