# Peer review of "Weathering and Material Characterization of ZTO/Ag/ZTO Coatings on Polyethylene Terephthalate Substrates for the Application of Flexible Transparent Conductors"

_coatings, doi:10.3390/coatings12091249_

Round 1
Reviewer 1 Report
The manuscript reported by Kao et al., is about the indoor accelerated and outdoor weathering tests to commercial OMO samples composed of ZTO/Ag/ZTO coating on PET substrates. I actually found the manuscript interesting and well-written. Thus, I recommend the manuscript for publication in Coating after some minor concerns as below have been addressed.
1. In row 226, the authors claimed that “The conductivity measurement indicates that the uniformity of the OMO samples is excellent…”. However, I actually could not find such conductivity measurement in the manuscript. Please verify this issue carefully.
2. In Figure 2, please perform some zoom-in XPS spectroscopy to be clear about the components of the materials. For example, please zoom-in to confirm the presence of Sn as well as the Ag substrate underneath.
3. In row 272 and Figure 4 of UV-vis measurements, the authors claimed that the outdoor and indoor give the same degradation level to the sample. But this is not convincing to me as the authors did not say anything about outdoor conditions, like sunlight, temperature, humidity, or moisture…. I believe that these conditions will strongly influence sample degradation. Thus, the authors should check this issue carefully.
4. Similar comment is also for the mechanical test in Figure 6, where the authors compared the mechanical properties between indoor and outdoor conditions. In Figure 6a, it seems that the sample at 23 days is brittle at less than 2 % strain compared to the sample tested under outdoor conditions where it is brittle at 2 % strain. Again, this result should be dependent on the outdoor conditions.
5. In the bending test, the movement speed of the clamps should play an essential role in the outcome of the results. Did the authors consider this issue in the experiment?
6. In row 353, the authors said that “the direction of the periodic cracks is always perpendicular to the direction of applied normal stresses…” However, from the SEM images (Figure 8 b and c), the cracks also occur in another direction. How do the authors comment on this issue?
Reviewer 2 Report
The paper “Weathering and Materials Characterization of ZTO/Ag/ZTO Coatings on Polyethylene Terephthalate Substrate for the Applications of Flexible Transparent Conductors” by Yu-Han Kao et al. presents some results obtained by the group about their investigation of aging effects on PET-OMO structure. The experiment was well designed and results are showed very well. I suggest the publication of the manuscript after minor revision. Some comments are given below for the improvement of the article.
Pag 2 line 47
Give an estimation of thickness threshold, or at least a range of thicknesses
Same at line 57, write the critical value or a range for it
Experimental methods
Write how was measure the sheet resistance and transmission briefly at the beginning of methods section, just a mention to the four-point method and spectrophotometer, that are better described after.
The pre-aging procedure, the annealing at 150°C, has a temperature above the glass transition temperature of PET, how this can affect the following measurement? If some measures were performed before the annealing would improve the manuscript show them briefly
